# Hepatitis Delta Virus Clade 8 Is the Predominant Clade Circulating in Botswana amongst People Living with HIV

**DOI:** 10.3390/v16101568

**Published:** 2024-10-04

**Authors:** Kabo Baruti, Wonderful T. Choga, Patience C. Motshosi, Bonolo B. Phinius, Basetsana Phakedi, Lynnette N. Bhebhe, Gorata G. A. Mpebe, Chanana D. Tsayang, Tsholofelo Ratsoma, Tendani Gaolathe, Mosepele Mosepele, Joseph Makhema, Roger Shapiro, Shahin Lockman, Sikhulile Moyo, Mosimanegape Jongman, Motswedi Anderson, Simani Gaseitsiwe

**Affiliations:** 1Research Laboratory, Botswana Harvard Health Partnership, Gaborone Private Bag BO 320, Botswana; kbaruti@bhp.org.bw (K.B.); wchoga@bhp.org.bw (W.T.C.); pmotshosi@bhp.org.bw (P.C.M.); bphakedi@bhp.org.bw (B.P.); lbhebhe@bhp.org.bw (L.N.B.); gmpebe@bhp.org.bw (G.G.A.M.); ctsayang@bhp.org.bw (C.D.T.); tratsoma@bhp.org.bw (T.R.); gaolathet@ub.ac.bw (T.G.); mosepele.mosepele@gmail.com (M.M.); jmakhema@bhp.org.bw (J.M.); rshapiro999@gmail.com (R.S.); shahinlockman@gmail.com (S.L.); smoyo@bhp.org.bw (S.M.); jongmanm@ub.ac.bw (M.J.); manderson@bhp.org.bw (M.A.); 2Department of Biological Sciences, Faculty of Science, University of Botswana, Gaborone Private Bag 00704, Botswana; 3School of Allied Health Professions, Faculty of Health Sciences, University of Botswana, Gaborone Private Bag 00704, Botswana; 4Faculty of Medicine, University of Botswana, Gaborone Private Bag UB 0022, Botswana; 5Department of Immunology and Infectious Diseases, Harvard T.H. Chan School of Public Health, Boston, MA 02115, USA; 6Department of Pathology, Division of Medical Virology, Stellenbosch University, Cape Town 7535, South Africa; 7School of Health Systems and Public Health, University of Pretoria, Pretoria 0002, South Africa; 8Africa Health Research Institute (AHRI), Durban 4013, South Africa; 9The Francis Crick Institute, London NW1 2BE, UK

**Keywords:** hepatitis delta virus, clades, mutations, Botswana

## Abstract

Hepatitis delta virus (HDV) co-infections more often result in severe hepatitis compared to hepatitis B virus (HBV) infections alone. Despite a high HDV prevalence (7.1%), information regarding circulating HDV clades is very limited in Botswana. We extracted total nucleic acid from confirmed HDV-positive samples and quantified their viral load. We then sequenced the large hepatitis delta antigen (L-HDAg) using Oxford Nanopore Technology (ONT). Genotyping was performed using the HDV Database, and HDV mutation profiling was performed on AliView. All participants with HBV genotypic information belonged to sub-genotype A1, and 80% (4/5) of them had a higher HDV viral load and a lower HBV viral load. We sequenced 75% (9/12) of the HDV-positive samples, which belonged to HDV clade 8. A total of 54 mutations were discovered, with the most prevalent being Q148R (16%), D149P (16%) and G151D (16%). Known mutations such as S117A, K131R, R139K and G151D were detected, while the other mutations were novel. Our results reveal that HDV clade 8 is the predominant clade in Botswana. The significance of all mutations remains unclear. Future studies with a larger sample size to detect other HDV clades that might be circulating in Botswana and functionally characterize the detected mutations are warranted.

## 1. Introduction

Hepatitis delta virus (HDV) is a defective, single-stranded ribonucleic acid (RNA) virus that requires envelope protein from a helper virus, the hepatitis B virus (HBV), for its replication [1]. The most severe type of viral hepatitis, caused by HBV/HDV infection, progresses rapidly towards hepatocellular carcinoma (HCC) and liver-related mortality [2]. A meta-analysis found that 12 million individuals worldwide are living with HDV, indicating that HDV is widespread globally [3]. The estimated prevalence of HDV in Africa is 5.97% among people who are chronically HBV-positive [3]. In the largest HDV study conducted in Botswana to date, we reported 7.1% HDV prevalence among hepatitis B surface antigen (HBsAg)-positive persons, which is higher than the estimated HDV prevalence in Africa [4].

The rolling circle amplification method has been used to amplify the 1.7 kilobase long, single-stranded circular RNA genome that makes up the HDV genome [5]. The hepatitis D antigen (HDAg), which is encased by the HBsAg, is linked to the HDV RNA in the nucleocapsid-like ribonucleoprotein (RNP) that makes up HDV virions [6]. An RNA editing event by the cellular adenosine deaminase (ADAR) results in the creation of the large HDAg (215 aa), while the unedited genome yields the small HDAg (196 aa) [7]. Eight different clades have been identified as a result of mutations caused by the HDV RNA polymerase’s inability to proofread [8]. These clades are associated with a distinct global and regional distribution, with clades 5–8 reported to be found in Africa [8].

It is important to determine circulating HDV clades and mutations, as they impact disease outcomes including progression to liver cancer and decompensation. Information regarding circulating HDV clades is very limited in African countries. Our aim was to identify the HDV clades and mutations that are currently in circulation in Botswana.

## 2. Materials and Methods

### 2.1. Anti-HDV Testing and Nucleic Acid Extraction

Previously confirmed HDV-positive samples *(n* = 12) were utilized in this study [4]. These confirmed HDV-positive samples had been screened for hepatitis B surface antigens (HBsAg) using the Murex HBsAg Version 3 Enzyme Linked Immunosorbent Assay (ELISA) kit (Murex Biotech, Dartford, UK), following the manufacturer’s protocol. HBV and HIV viral loads were quantified among HBsAg-positive samples using the Roche COBAS Ampliprep/Taqman Analyzer (Roche Diagnostics, Mannheim, Germany) in a previous study [9]. Next-generation and Sanger sequencing were used to generate near full-length HIV-1C sequences through a collaboration between the Biopolymers Facility at Harvard Medical School and the PANGEA consortium, in a previous study [10].

The samples were screened for anti-HDV at least once using the General Biologicals HDV Ab kit (General Biologicals Corporation, Taiwan, China), following the manufacturer’s instructions. The Daan Gene nucleic acid extraction kit (Daan Gene Co, Ltd., Guangzhou, China) was used to extract total nucleic acid (TNA) from confirmed positive anti-HDV samples, following the manufacturer’s instructions. This kit required a 200 μL sample volume and eluted 50 μL of the extract that was stored in small aliquots in a freezer at −20 °C temperature.

### 2.2. HDV RNA Load Quantification and Next-Generation Sequencing

HDV viral RNA was previously detected using the Altona Diagnostic RealStar^®^ HDV RT-PCR 1.0 detection kit (Altona Diagnostics, Hamburg, Germany), with a few minor adjustments to the manufacturer’s instructions [4]. The total reaction volume used was 25 μL instead of 50 μL due to the limited eluted TNA volume. Residual RNA from viral load quantification testing was used for complementary DNA (cDNA) synthesis using the LunaScript RT master mix (New England BioLabs, Ipswich, MA, USA). LunaScript RT SuperMix (2 µL) was added to 8 µL of the extracted sample in a 96-well plate. The plate was sealed, spun down and incubated in a thermal cycler using the following conditions: 25 °C for 2 min, 55 °C for 10 min, 95 °C for 1 min and a hold stage at 4 °C.

This step was followed by tilling PCR using the rapid barcoding and midnight RT-PCR expansion (SQK-RBK110.96 and EXP-MRT001) kits, with minor modifications. HDV tilling primers (pool A and B) were manufactured by Inqaba Biotech (Pretoria, South Africa) and added to the reaction mix instead of the random primers provided in the kit. Other modifications included the addition of a 4 μL HDV RNA template volume and increasing the PCR cycles to 40. The reaction mixes for each primer pool consisted of 3.7 µL of nuclease-free water and 6.25 µL of Q5 HS Master Mix and HDV primer pool A/B, resulting in a total reaction volume of 10 µL. Using a multichannel pipette, 10 µL of HDV pools A and B was aliquoted into a clean 96-well plate, and 2.5 μL of each RT reaction was transferred to the corresponding well for both HDV pools A and B in the PCR plate. The plate was sealed, spun down briefly and incubated using the following conditions: 98 °C initial denaturation for 30 s, 98 °C denaturation for 15 s, 65 °C annealing and extension for 5 min (40 cycles) and a hold stage at 4 °C.

Barcodes from the Oxford Nanopore Rapid Barcoding kit were added to the amplicons and purified using AMPure beads, according to the manufacturer’s protocol. Briefly, 2.5 μL aliquots of each sample were transferred from the Rapid Barcode Plate to the corresponding well of the Barcode Attachment Plate, mixed by pipetting. The Barcode Attachment Plate was sealed, spun down and incubated in a thermal cycler at 30 °C for 2 min and then at 80 °C for 2 min. The Elution Buffer (EB) was added to the DNA library (800 ng) to make up the volume to 11 µL. The Rapid Adapter F (1 µL) was added to 11 µL of barcoded DNA and incubated at room temperature for 5 min. After the bead clean-up, the library was loaded on a prepared GridION X5 R9.4.1 flow cell, and sequencing was performed using MinKNOW software with high-accuracy basecalling using Guppy.

### 2.3. Sequence Sorting and Phylogenetic Analysis

Raw FastQ files obtained from the GridION were uploaded into the Genome Detective tool to perform reference-based assembly and remove adapters and poor-quality reads [11]. A phylogenetic tree was constructed using iqtree2 and after choosing the best fitting model. A dataset containing 11 HDV clade 8 reference sequences (accession numbers: MT138421, ON286993, OK349702, LT594488, LT604974, AM183327, LS482960, LS482966, AJ583882, AM183330 and AJ584844) were downloaded from GenBank and incorporated in the alignment containing BW sequences. Tree visualization and annotations were performed in FigTree v1.4.3.

### 2.4. Mutational Analysis

Obtained FASTA files were genotyped simultaneously using National Centre for Biotechnology Information (NCBI) and Hepatitis D Virus (HDV) Database (http://hdvdb.bio.wzw.tum.de/ accessed on 15 July 2024). In addition, a dataset containing seven (7) HDV genotype 8 reference sequences (accession numbers: ADR56049, CAE51164, CAJ66091, CAJ66094, SBT96884, SCC98323 and SCC98324) was downloaded from GenBank and incorporated in the alignment in Aliview. We positioned sequences relative to the protein length, and L-HDAg mutations were manually profiled from the aligned files. HBV genotypes were determined using Geno2pheno version 2.0 (https://hbv.geno2pheno.org) (last accessed 11 December 2023) and confirmed using the Stanford HBV sequence database.

## 3. Results

### 3.1. HDV Sequencing

We analyzed 15 anti-HDV-positive samples from the most recent and largest HDV study in Botswana to date, which utilized samples from a total of 211 participants with HBV [4]. The parent study generated 15 anti-HDV-positive samples that were eligible for HDV sequence analysis. Of these, only 12 samples with sufficient volume were sequenced. We successfully sequenced 75% (9/12) samples, which all belonged to HDV clade 8. All participants with HBV genotype information (*n* = 5) belonged to sub-genotype A1, resulting in an HBV-A1/HDV-8 genotype combination. Most of these participants (80%) had inversely proportional HDV/HBV viral load (a higher HDV viral load and a lower HBV viral load) (Table 1).

### 3.2. HDV Mutation Analysis

We then performed an L-HDAg amino acid sequence alignment of seven (7) successfully sequenced isolates with HDV clade 8 reference sequences to determine amino acid substitutions (Figure 1). The amino acids in our samples covered only part of the L-HDAg (aa 113–214). No indels were observed, and there was no variability in the leucine position 115 among our sequences.

Mutations were defined as amino acid substitutions in the L-HDAg of our sequences relative to the amino acids in the aligned reference sequences. A total of 54 aa mutations were identified in this region, with the most prevalent being the K131R (16%), Q148R (16%), D149P (16%) and G151D (16%) mutations (Figure 2). Previously identified mutations such as S117A, K131R, R139K and G151D were detected while the rest of the mutations were not found in the available literature. The clinical significance of these mutations remains unknown.

### 3.3. Phylogenetic Tree Analysis

HDV sequences were identified using the BLAST tool that based nucleotide similarity on NCBI. After results from this identification revealed that all isolates belonged to HDV clade 8, a maximum likelihood (ML) phylogenetic tree was drawn to determine the evolutionary relatedness with sequences from the HDV database. The ML tree demonstrated clustering of our sequences with each other and evolutionary relatedness with HDV clade 8 reference sequences from Angola and Namibia, which are neighboring countries to Botswana (Figure 3).

## 4. Discussion

Limited data on circulating HDV clades are available in African countries, including Botswana. To the best of our knowledge, this is the first study to reveal circulating HDV clades in Botswana. All samples that were successfully sequenced were identified as HDV clade 8. This finding correlates with previous studies reporting the detection of HDV clade 8 predominantly in Africa, especially in West African countries such as Gabon, Congo, Senegal and Ivory Coast [12]. Despite limited published data on the circulating clades in Southern Africa, we found HDV clade 8 sequences from neighboring countries, Angola (OK349702) and Namibia (MT138421), in GenBank. Our findings demonstrate that Botswana is the third country in this region where this clade has been detected, and it may represent the predominant circulating clade in Southern Africa.

There is limited information on the interaction of HDV clade 8 with HBV sub-genotype A1, which is the most predominant HBV genotype in Botswana. It is important to determine HBV-HDV genotype combinations because they are associated with varying disease outcomes. A previous study reported adverse outcomes among chronic HBV-positive patients with HBV genotype C and HDV clade 1. Furthermore, different HBV genotypes have variations within the reverse transcriptase region, which may result in the development of antiviral drug resistance. We recorded a higher HDV viral load and a lower HBV viral load in most participants (80%), which is in concordance with the literature and suggests that HDV tends to dominate HBV [13]. HDV is reported to have the capability to inhibit HBV activity by suppressing the HBV enhancer, allowing HDV to benefit more from HBsAg for its replication [13]. Additionally, it is important to note that these participants were on lamivudine and tenofovir therapy, both of which have anti-HBV activity. However, they were not receiving any specific treatment for HDV as it was not included as part of the national viral hepatitis treatment program when the participant samples were collected. There are conflicting data from longitudinal studies regarding the efficacy of tenofovir on HDV viral load, while lamivudine information is very limited. One study reported significantly reduced serum HDV-RNA in HIV-infected patients with HDV after treatment with tenofovir [14], while another study reported that tenofovir alone is not sufficient for the effective resolution of HDV infection as none of the HIV-HBV-infected participants achieved undetectable HDV viral load [15]. Therefore, the ability of tenofovir to reduce the HBV viral load to a greater extent than HDV viral load could have contributed to the inversely proportional viral loads observed in our results. We were able to genotype from one participant with undetectable HDV viral load. It was possible to perform HDV genotyping on this participant because the Altona Diagnsotics RealStar HDV real-time PCR assay has a high lower limit of detection. Therefore, the target not detected (TND) means that the HDV RNA load was below the lowest detectable viral load by the kit.

A frequent characteristic of HDV is a very significant variability in the amino acid composition of the HDAg protein [16]. However, the functional implications of clinically recognized alterations in this protein’s amino acid composition remain unclear. While it has been reported that two arginine-rich motifs (residues 97 to 107 and 136 to 146) are necessary to form protein complexes with HDV RNA [17], a recent study on the RNA binding activity of HDAg suggested that the N-terminal domain and a wide range of basic and non-basic residues involved in this reaction over the whole protein sequence are primarily responsible for this reaction [18]. Our sequences could not cover the conserved arginine-rich region between amino acids 2–27 (the N-terminal region) and 97–107 [17]. However, the third proposed RNA-binding domain (136–146) contained the R139K mutation (lysine substitutes arginine) and E136D (aspartic acid substitutes glutamic acid). Although the R139K mutation has been previously detected, its role is not well understood [19]. However, the presence of lysine instead of arginine might influence RNA binding inside this domain or imply a comparable function for lysine. Even though mutations S117A, K131R and G151D have been detected in Central African Republic and Iran (S117A), their functions are yet to be characterized [20,21]. Functionally important amino acid positions such as leucine-115, which is necessary for in vitro RNA binding activity and viral replication [22], showed no variability, notwithstanding studies reporting high amino acid variability in the HDAg protein [16].

After identifying that all isolates belonged to HDV clade 8, a maximum-likelihood (ML) phylogenetic tree was drawn to determine the evolutionary relatedness with sequences from the NCBI database (Figure 3). The ML tree demonstrated evolutionary relatedness with HDV clade 8 reference sequences from neighboring countries, Angola (OK349702) and Namibia (MT138421). HDV sequences from West Africa also clustered among themselves so this may suggest similar transmission dynamics among countries from the same region. Our study was limited by the few anti-HDV-positive samples that were available, which affected the downstream analysis of our data. This means that genotyping was based on a few samples; therefore, the possibility of missing other circulating HDV clades and mutations cannot be ruled out. However, this the first study in Botswana sequence HDV, therefore, it provides a good basis for further investigation, given the importance of HDV-HBV infections towards viral hepatitis elimination. Furthermore, there was a lack of HBV genotype data for some of the HDV positive participants due to failure to amplify associated in some cases with low HBV viral loads. This might have led to HBV primers not working with our template, hence failure of amplification. Other participants had insufficient sample volumes; therefore, we could not perform HBV genotyping to estimate the potential effect of the HBV-HDV genotype combinations on the HBV and HDV viral loads. Most patients with HDV sequence data come from the same village (Gumare) and therefore might have similar risk factors and transmission dynamics; as such, this information cannot be generalized on the clade circulating throughout Botswana. We did not use other enzymes besides the Lunascript RT master mix during the cDNA synthesis step; therefore, we could not rule out that some of the mutations observed could have been introduced by the enzyme.

## 5. Conclusions

Our results demonstrate that HDV clade 8 is the predominant genotype circulating in Botswana. Future studies should use a larger sample size and whole genome sequencing of more HDV-positive samples to detect more mutations and other HDV clades that might be circulating in Botswana. There is an ongoing need to perform functional characterization of the identified mutations to determine their clinical significance.

## Figures and Tables

**Figure 1 viruses-16-01568-f001:**
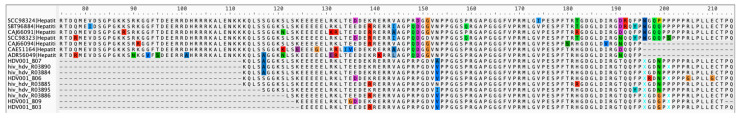
L-HDAg amino acid alignment of the isolates.

**Figure 2 viruses-16-01568-f002:**
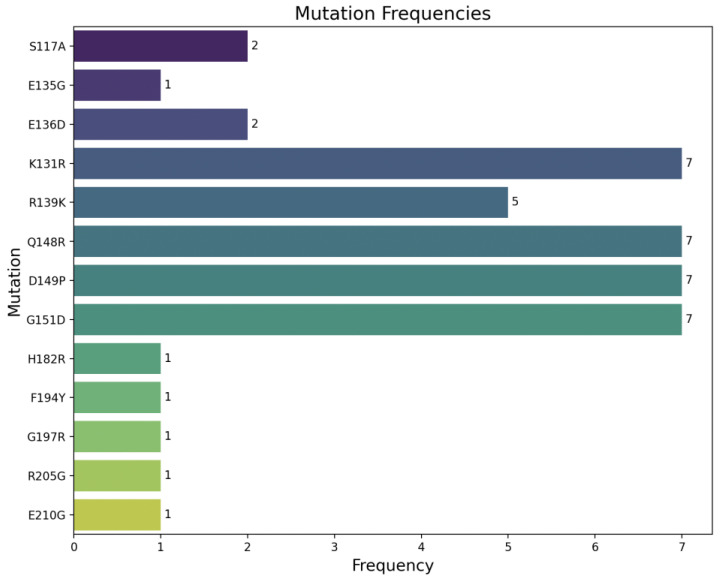
Frequency of HDV mutations.

**Figure 3 viruses-16-01568-f003:**
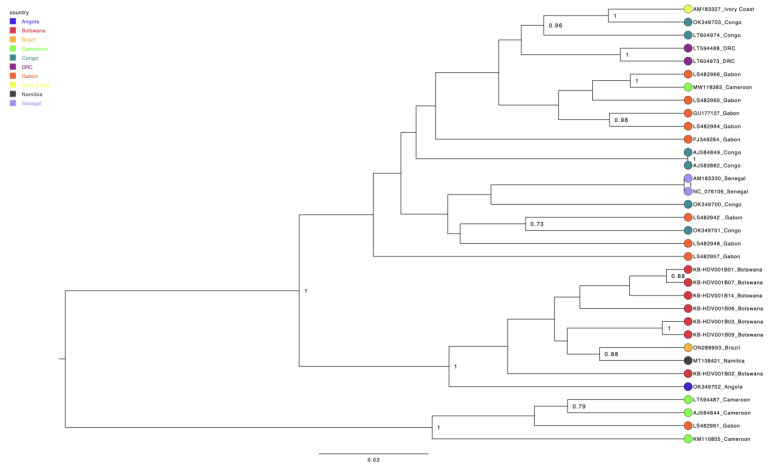
HDV maximum likelihood phylogenetic tree.

**Table 1 viruses-16-01568-t001:** Participants with HDV sequences.

ID	Sex	Age	Village	HBV Genotype	HDV Clade	HDV Viral Load	HBV Viral Load	HIV Viral Load	cART Regimen
1	F	36	Gumare	A1	Clade 8	417,496	41	N/A	3TC + NVP + ZDV
2	M	45	Gumare	A1	Clade 8	172,871	<20	N/A	FTC + LPV + TDF
3	F	40	Gumare	A1	Clade 8	2,569,776	481	N/A	EFV + FTC + TDF
4	M	51	Gumare	A1	Clade 8	13,127	TND	N/A	3TC + EFV + ZDV
6	M	57	Rakops	N/A	Clade 8	148,078	36	40	N/A
7	M	58	Gumare	N/A	Clade 8	93,696	<20	40	3TC + EFV + ZDV
8	F	38	Gumare	N/A	Clade 8	2090	N/A	40	3TC + NVP + ZDV
11	F	35	Rakops	N/A	Clade 8	72	>170,000,000	21,574	FTC + TDF
14	F	33	Shakawe	A1	Clade 8	TND	1399	N/A	3TC + NVP + ZDV

ID—identification; M—male; F—female; TND—target not detected; N/A—information not available; HDV—hepatitis delta virus; HBV—hepatitis B virus; HIV—human immunodeficiency virus; cART—combination antiretroviral; 3TC—lamivudine; NVP—nevirapine; ZDV—zidovudine; FTC—emtricitabine; LPV—lopinavir; TDF—tenofovir; EFV—efavirenz.

## Data Availability

The data presented in this study are available from the corresponding author upon request. The data are not publicly available as the sequences are currently being analyzed for other objectives of the bigger project.

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
