# Peer review of "Hepatitis Delta Virus Clade 8 Is the Predominant Clade Circulating in Botswana amongst People Living with HIV"

_viruses, 2024, doi:10.3390/v16101568_

Round 1

Reviewer 1 Report

Comments and Suggestions for Authors Dr Baruti and colleagues carried out a work on 9 HIV HBV HDV co-infected patients, analyzed very well from the point of view of mutations and virological research. Furthermore, this paper is very interesting regarding an initial and preliminary descriptive work of prevalence of clade 8 in Botswana. On the other hand, the paper has many limitations: It's only mentioned once that the patients are HIV. The HIV co-infection data must be included in the methods and in the text extensively. In the conclusions it should be made much clearer that the work has many limitations as the majority of patients (6/9) come from the same village. who are all HIV patients (same risk factors? same family? any information on transmission?). that this information cannot be generalized on the genotype circulating throughout Botswana (based on only 9 patients...). The title should be changed, such as "Preliminary results...in HIV HDV patients... ". Furthermore, from Table 1, is patient 11 taking antiretroviral therapy correctly or does he have resistance?? Also, how was it possible to perform the HDV genotype test if the Delta viral load was negative in patient 14?? Are there therefore 8 patients to consider for the paper? Is there any possibility to expand the HDV genotype determinations on others HBV samples from others villages?

Comments on the Quality of English Language

na

Reviewer 2 Report

Comments and Suggestions for Authors

The manuscript from Baruti et al. investigate the circulating HDV clades in Botswana, they found that HDV clade8 is most predominant clade. They used Oxford Nanopore sequenced the large hepatitis delta antigen (L-HDAg). They found multiple mutations and sub-genotype A1, and 80% (4/5) of them had a higher HDV viral load and a lower HBV viral load. The references are cited properly. The tables in the current manuscript are clear and support the author’s points. The work appears to carefully conducted, and the results will be well received by Hepatitis Virus community. However, I do have some questions/concerns that I think the authors should address before publication.

1.       In current manuscript, the authors measure the mutation by nanopore, which requires reverse transcription, here the authors used LunaScript RT master mix. However, the reverse transcriptase has non-specific mutation, is the mutations found in current study comes from non-specific mutation or from HDV RNA? To exclude this factor, the author should use other enzymes side by side.

2.       The authors should provide more details about nanopore sequencing methods.

3.       In line 75, 76 and 82, XXµL should be XX µL.

4.       The authors should provide more references, for example, doi: 10.3390/ijms232415973

Comments on the Quality of English Language

English is fine, only minor editing of english is required.  

Round 2

Reviewer 1 Report

Comments and Suggestions for Authors

Thank you for considering HIV in the title and to have emphatized the limits of the study in the discussion 

Comments on the Quality of English Language

Na